# CD99: A Key Regulator in Immune Response and Tumor Microenvironment

**DOI:** 10.3390/biom15050632

**Published:** 2025-04-28

**Authors:** Maria Cristina Manara, Valentina Fiori, Angelo Sparti, Katia Scotlandi

**Affiliations:** 1Experimental Oncology Laboratory, IRCCS Istituto Ortopedico Rizzoli, 40136 Bologna, Italy; mariacristina.manara@ior.it; 2Diatheva srl, 61030 Cartoceto, Italy; v.fiori@diatheva.com; 3Department of Biomolecular Sciences, University of Urbino Carlo Bo, 61029 Urbino, Italy; a.sparti@campus.uniurb.it

**Keywords:** CD99, Ewing sarcoma, osteosarcoma, glioblastoma, leukemia, immunotherapy, targeted therapy, tumor microenvironment, immune response, immune evasion

## Abstract

CD99 is a membrane protein critical for various immunological functions, including T-cell activation, protein trafficking, cell apoptosis, and leukocyte movement. It is also highly expressed in certain malignant tumors, contributing to the development, invasion, immune evasion, and adaptation of tumor cells to stress stimuli, including drug resistance. CD99 is crucial at the intersection of normal biological processes and pathological conditions like cancer. While research indicates that CD99 may interact homotypically, there is evidence of some heterotypic ligands that align with its roles. The development of multiple anti-CD99 antibodies has shed light on its functions, particularly regarding interactions between tumor cells that overexpress CD99 and immune cells expressing the same protein within the microenvironment. Anti-CD99 antibodies effectively eliminate tumors and attract immune cells to the tumor area. Additionally, CD99 influences the expression of specific immune checkpoint molecules, such as CD47, paving the way for potential combinations of anti-CD99 with immune checkpoint inhibitors. This review explores CD99’s role in normal physiology and cancer biology, focusing on how monoclonal antibodies affect CD99 expression and activity, thereby influencing cancer cells’ interactions with their microenvironment. It summarizes key findings about how these changes impact cancer cell behavior and the effectiveness of treatments.

## 1. Introduction

CD99 antigen, encoded by the *MIC2* gene, is a 32 kDa transmembrane protein widely expressed in nearly all human cell types, particularly in hematopoietic cells, the thymus, endothelial cells, Sertoli cells, and cancer cells [1].

Over the years, more evidence has highlighted the dual role of CD99 in immune regulation and tumor progression, suggesting a link between these two functions in the tumor microenvironment (TME).

In physiological conditions, this molecule is implicated in the regulation of several crucial immunological functions, such as the proliferation and stimulation of mature peripheral T cells, the transport regulation of Major Histocompatibility Complex (MHC) class I molecules from the Golgi complex to the cell surface, T-cell migration, and the diapedesis of monocytes across the endothelium [1]. Furthermore, CD99 plays a role in lymphocyte development, as CD99-deficient fetuses have shown significant impairment in thymic development, indicating a role for CD99 in normal thymus ontogeny [2]. CD99 has been shown to elicit homotypic cell aggregation and induce cell death at critical stages of thymocyte differentiation [3,4]. Various CD99 targeting ligands have been developed as recombinant proteins and monoclonal antibodies, which upregulate Interleukin-6 (IL6), Tumor Necrosis Factor alpha (TNFα) [5,6], Interferon-gamma (IFNγ), and Type 1 T helper (Th1) [7].

Recent studies have highlighted how the triggering of CD99 on M0/M2-like macrophages promotes their polarization toward an inflammatory M1-like phenotype, contributing to tumor killing [8,9]. All these data provide evidence of the potential to tune the tumor microenvironment.

In pathological conditions, increased CD99 expression has been associated with several malignancies, including Ewing sarcoma (EWS) [10], myeloid malignancies [11,12,13], lymphoblastic lymphoma/leukemia [14], and malignant glioma [15,16]. In these instances, CD99 is recognized as an oncogene that enhances cell migration and invasion, leading to tumor cells exhibiting higher metastatic potential. Regarding acute myeloid leukemia (AML) and myelodysplastic syndromes (MDS), higher expression of CD99 has been linked to leukemia-initiating cells compared to normal hematopoietic cells [11].

As a result, CD99 has been used as a biomarker and a promising therapeutic target for tumors that overexpress CD99. Due to its immune regulatory function, triggering CD99 may act as an immunomodulator, boosting the activity of immune cells and possibly serving as an adjuvant alongside other tumor therapies when combined with anti-CD99 antibodies.

Key points

CD99 is expressed in normal cells, particularly immune cells, and in some tumors.CD99 regulates various biological processes such as adhesion, transendothelial migration, differentiation, and cell death, thereby affecting immune function, inflammation, and cancer metastasis.The CD99 molecule is found at the interface between immune cells and tumors and may play a dual role as a modulator against CD99-expressing tumor cells and immune cells.

This review examines the role of CD99 in various aspects of normal physiology and cancer biology, focusing on its functions in immune cells. It discusses how monoclonal antibodies can affect CD99 expression and activity, influencing homotypic and heterotypic interactions between cancer cells and their microenvironment. Additionally, it provides an overview of the key findings regarding how these modifications can impact cancer cell behavior and contribute to the effectiveness of cancer treatment protocols.

## 2. CD99 Antigen

CD99 is encoded by the pseudoautosomal gene *MIC2*, which is located in the pseudoautosomal region (PAR) of both the X (Xp22.33-Xpter) and Y (Yp11-Ypter) chromosomes in humans [17,18,19,20,21,22]. For a detailed description of the CD99 gene/protein structure, refer to Pasello et al. 2018 [1].

The *MIC2* gene produces two proteins through alternative splicing: the full-length CD99 (type I, CD99wt), with 185 amino acids and 32 kDa, and the truncated CD99 (type II, CD99sh), containing 161 amino acids and 28 kDa [23]. The CD99sh transcript has an 18-base pair insertion at the boundary of exons 8 and 9, leading to an in-frame stop codon, creating a truncated polypeptide [23].

Due to the alternative splicing process, the potential phosphorylation sites for serine at amino acid position 168 and threonine at position 181 are absent in CD99sh. The two CD99 isoforms are expressed in a cell type-specific manner and regulate different functions [1]. CD99 can form homodimers through its extracellular domain. Additionally, when stimulated, the two CD99 isoforms naturally undergo heterodimerization on cell surfaces and act as receptors [24].

Few studies have specifically investigated the expression of these two isoforms in various cell types and their impact on CD99-mediated intracellular pathways. Unless stated otherwise, this review primarily focuses on data related to CD99wt.

The search for CD99 homologs has only been successful in primates, indicating a high level of sequence divergence of this gene during evolution [25].

Suh and co-authors described the presence of a CD99 paralogous mouse gene, CD99 antigen-like protein 2 (CD99L2), and its orthologs in humans [26]. For details regarding the roles and mechanisms, please refer to Pasello et al. [1].

## 3. CD99 Ligands

While the functions of CD99 are established, its ligands and downstream signaling pathways remain unclear. Since CD99 is expressed in certain pathological conditions, such as tumors and a subset of immune cells, recent research has focused on the interactions between tumor cells and TME. This has contributed to a clearer definition of the ligands and downstream effectors. Most publications indicate that a homotypic interaction between CD99 molecules is the ligand for CD99 found in various cell types. Nevertheless, in certain instances, a heterotypic interaction has been observed [27].

Recently, a dimer form of a soluble human CD99HIgG fusion protein was produced to study the interaction between CD99 and its ligands expressed in cells. It was demonstrated that CD99 ligands were present on the surfaces of monocytes, natural killer (NK) cells, and dendritic cells but not on those of B lymphocytes and resting T lymphocytes. Furthermore, this interaction regulates the production of proinflammatory cytokines IL-6 and TNF-α in NK cells, monocytes, and activated T cells [28,29].

In melanoma, a tumor that expresses high levels of CD99 [30], an integrated analysis of single-cell and bulk RNA data revealed that, most significantly, melanocytes communicate with cancer-associated fibroblasts (CAFs), endothelial cells, NK cells, and T cells, as well as with macrophages. Specifically, interactions occurred between NK cells and T cells via CD99-CD99, while the interaction between melanocytes and macrophages involved communication through CD99-CD99L2 [31].

A study on lung tumors that metastasize to the brain found higher transcriptomic changes in brain metastases than in lymph node metastases and primary tumors. Several pathways linked to cell surface, adhesion, and migration—including CD99—showed a significant decrease in brain metastases. The research highlighted specific ligand–receptor pairs that are common to both lymph node and brain metastases, particularly noting the interaction between Secreted Phosphoprotein 1 (SPP1) and CD99 in T cells [32].

A recent study by Zhao on breast cancer revealed significant heterogeneity in cell communication pathways that influence macrophage infiltration. It highlighted an interaction between luminal cancer cells and myeloid cells, such as monocytes, macrophages, and dendritic cells. The luminal subtype demonstrated CD99 signaling activation, which is absent in the genomic stable subtype. The analysis identified a high probability of communication between macrophage migration inhibitory factor (MIF), CD99, and paired Ig-like type 2 receptor (PILRa), suggesting a regulatory role in macrophage polarization. A notable correlation was observed between CD99 expression and M2 macrophage markers like peroxisome proliferator-activated receptor gamma (PPARG), Arginase 1 (ARG1), Transforming growth factor beta 1 (TGFB1), and Macrophage receptor with collagenous structure (MARCO), possibly regulated by ligand–receptor interactions involving CD99. This study emphasizes the regulatory roles of CD99 and PILRa in the communication between luminal cancer cells and macrophages in breast cancer. There is an ever-growing body of evidence suggesting that the CD99:PILRa/b interactions significantly affect the quality of the innate immune response [33].

Similar findings were also observed in EWS cell communication with macrophages in the tumor niche, as will be discussed further in this review in the section dedicated to EWS.

The PILRs are known heterophilic binding partners of CD99. They have been described as lectins on NK cells and dendritic cells, which bind to O-linked carbohydrates of CD99 on lymphocytic cells and activate their cytotoxic capacity [34]. Goswami and colleagues demonstrated that the binding of endothelial cells to PILRs was completely abolished by antibodies against CD99 or by silencing CD99 expression using small-interfering RNA, thus proving this interaction is crucial for diapedesis of leukocytes (see below) [35].

In EWS cells, a recent study identified Growth and Differentiation Factor 6 (GDF6) as a ligand of CD99. GDF6 is a member of the bone morphogenetic protein (BMP) family of cytokines, which plays a crucial role in sustaining Ewing sarcoma proliferation by inhibiting Src hyperactivation. This study demonstrated that the interaction between the GDF6 protodomain and the extracellular domain of CD99 facilitates the recruitment of C-terminal Src kinase (CSK) to the YQKKK motif located in the intracellular domain of CD99, thereby inhibiting Src activity. The knockdown of GDF6 induced hyperactivation of Src and p21-dependent growth arrest [36].

## 4. CD99 Functions in Immune Cells

### 4.1. The Role of CD99 in Immune T Cells

Although the two CD99 isoforms have been reported to regulate different functional processes [6,37,38], very few studies have demonstrated how these isoforms are expressed in specific cellular environments and how they influence CD99-mediated intracellular pathways. The two CD99 isoforms may exist on the surface of T cells either as a homodimer of the long form (type I) or as a heterodimer composed of both the long and short forms [23]. These different forms of CD99 are linked to T-cell maturation, with heterodimeric variants typically expressed together in early-stage double-positive CD4 CD8 thymocytes, while CD99 type I is expressed in late-stage thymocytes and peripheral mature T cells [37].

In the thymus, CD99 has been shown to trigger homotypic cell aggregation and cell death during crucial stages of thymocyte differentiation, when positive selection is known to occur [3,4,39]. Specifically, CD99 has been demonstrated to induce cell death in immature CD4^+^ CD8^+^ thymocytes with intermediate CD3 density, including all detectable CD69^+^ cells, while not affecting the survival of other thymocytes or T cells [3,4,39]. When expressed, the two isoforms form covalent heterodimers in glycosphingolipid rafts, inducing sphingomyelin degradation. Cholesterol depletion experiments showed that this localization is essential for apoptosis induction. Thus, CD99 isoform expression patterns define T-cell functions [37].

CD99 engagement has been shown to expose phosphatidylserine on the surface of immature thymocytes [40,41]. The cell death mechanism depends on the specific CD99 domains activated by distinct antibodies, with cell death occurring through either classical or non-classical apoptotic pathways [39,40]. The underlying mechanism by which distinct CD99 domains trigger different death pathways remains to be elucidated. However, a similarly intricate scenario, where the engagement of different domains induces either caspase-dependent or --independent cell death, has been documented in the context of other molecules, such as MHC class I molecules [42,43].

Choi et al. reported that the engagement of CD99 with antibodies resulted in an upregulation of T-cell receptors (TCR), MHC I, and MHC II molecules on the surface of thymocytes, particularly at the plasma membrane and at cell-cell contact sites, enhancing the CD99-mediated interaction between T cells [41,44]. This increase results from improved molecule mobilization to the plasma membrane, rather than from increased RNA and protein synthesis. CD99 plays a role later in the protein transport process, particularly during trafficking from the trans-Golgi network (TGN) to the plasma membrane, by activating cytoskeletal components or linker proteins, which induce accelerated mobilization of specific antigens to the cell surface. This effect is particularly significant in TCR-low subpopulations of immature double-positive thymocytes. These findings suggest that CD99-dependent upregulation might have implications for positive selection during thymocyte ontogenesis [41].

As demonstrated in the study by Brémond et al., the regulation of Human leukocyte antigen (HLA) class I surface expression requires the interaction between CD99 and p230/golgin-245. This interaction is essential for the trafficking of MHC I from the TGN to the cell surface [44,45]. The engagement of MHC class II molecules by specific antibodies exerts an antagonistic effect on CD99 engagement-related phenotypes, with an inhibition of CD99-induced upregulation of TCR [46], indicating that complex regulatory interactions exist between CD99 and MHC class I and II signaling during the development and maturation of thymocytes [46]. It is hypothesized that the antagonistic effect of MHC class II engagement on CD99-related phenotypes results from the idea that the signals delivered by MHC class II engagement might converge on the microtubule network with those transduced by CD99 engagement, thereby blocking the CD99-induced accelerated transport of TCR and MHC molecules. Thus, immature thymocytes’ positive selection and maturation may result from a balance between CD99 and MHC class II signaling, enhancing positive selection efficacy [46].

Yoon et al. proposed CD99 as a T-cell receptor co-stimulatory molecule. Upon triggering, CD99 engagement upregulated CD3, Lymphocyte function-associated antigen 1 (LFA-1), Intercellular Adhesion Molecule 1 (ICAM-1), and CD5 in Jurkat T cells. Similarly to MHC I and II, this increase is primarily due to the translocation of these molecules from the cytoplasm to the plasma membrane rather than increased RNA and protein synthesis [47]. CD99 stimulation with antibodies boosted the proliferation of human CD4 T cells. Crosslinking CD99 with anti-CD99 monoclonal antibodies (mAbs) and anti-CD3 Ab increased the expression of CD25, CD69, and CD40L, indicating T-cell activation [6] (Figure 1). T-cell activation may trigger exocytic and endocytic events. Yoon et al. demonstrated that CD99 plays a role in these processes. It has been shown that CD99 affects the trafficking of many cell surface molecules including the T-cell receptor complex. CD99 engagement led to export of recycling TCR-CD3 complexes and clustering of lipid rafts and of the actin cytoskeleton [48]. CD99 is upregulated in memory T and B cells, which express exocytic molecules stored in endosomes or secretory lysosomes [49,50].

Thus, CD99’s function could go beyond being simply a costimulatory molecule on T-cell surfaces, indicating a possible involvement in effector functions like target cell lysis by cytotoxic T cells or NK cells.

Mechanistically, ligation of CD99 led to tyrosine phosphorylation of a 29-kDa protein, suggesting a CD99-induced signal transduction pathway [6,7]. This prompted differential activation of members of the mitogen-activated protein kinase (MAPK) family, including extracellular signal-regulated kinase (ERK), c-Jun N-terminal kinases (JNK), p38, mitogen-activated protein kinase (MAPK), and src kinase [51,52]. The interaction between CD99 and suboptimal CD3-induced T-cell activation resulted in the translocation of TCR complexes to lipid rafts, thereby enhancing TCR-mediated signaling [53]. Upon T-cell activation, CD99 translocates to immunological synapses, and anti-CD99 mAb hasbeen shown to inhibit T-cell proliferation, indicating the significant role of CD99 in T-cell activation [54].

### 4.2. The Role of CD99 in Immune B Cells

CD99 may also play a key role in the early stages of B lymphopoiesis. CD99 expression has been strongly correlated with the maturation of normal B-cell precursors, exhibiting the highest levels observed in the most immature B-cell precursor (BCP) stages [55]. In these immature precursors, CD99 triggering with anti-CD99 antibodies can induce cell death after long-term incubation, suggesting a role for CD99 in clonal selection. The alternatively spliced CD99 type II mRNA is either absent in normal BCPs or present at extremely low levels and does not affect maturation—CD99 appears to be critically involved in regulating the cell cycle and maintaining differentiation in mature B cells. In B cells, the short isoform of CD99 inhibits homotypic adhesion, whereas the long isoform enhances cell-cell adhesion. The divergent effects of CD99 isoforms on homotypic B cell aggregation stem from their opposing roles in regulating the expression of the cell adhesion molecule LFA-1 [23].

Among B cell subsets in the tonsils, CD99 expression was highest in plasma cells (PCs). CD99 expression increased during in vitro differentiation within the germinal center of B cells, reaching its peak in PCs. CD99 engagement did not influence apoptosis, differentiation, or antibody secretion of PCs, but it diminished their chemotactic migration toward CXC motif chemokine 12 (CXCL12) and reduced ERK activation by CXCL12 [56] (Figure 2).

### 4.3. The Role of CD99 in Other Immune Cells

In the bone marrow, CD34+ cells display a CD99high population with greater migratory potential than the CD99low population. Early and late Colony-Forming-Unit-Erythroid (CFU-E) cells in vivo are also characterized by CD99high and CD99low expression. Inhibition of CD99 decreases the proliferation of pluripotent stem cell-derived cells and promotes erythroid maturation. The CD99high subpopulation in Pluripotent Stem Cells (PSC) and UCB (umbilical cord blood)-CD34+ cell-derived endothelial progenitor cells (EPCs) derived from CD34-positive HUVECs both generate granulocyte-macrophage progenitor cells while burst-forming unit-erythroid (BFU-Es) cells are primarily found in the CD99low subpopulation, suggesting that the CD99high subcluster retains the potential for granulocyte, macrophage, and erythroid cell differentiation. This also indicates that the CD99high subpopulation expresses these progenitor cells early. Treatment with CD99 antagonists like clofarabine and 2-chlorodeoxyadenosine inhibited cell proliferation, increased Baso-E and Poly-E cells, and decreased Pro-E cells. This indicates that CD99 is a marker for the proliferating subpopulation and plays a role in erythroid cell maturation. Additionally, the authors found that macrophages were linked to ex vivo erythropoiesis through CD99 expression level, establishing a CD99-CD99 contact with specialized erythroid cells. This highlights a novel mechanism through which macrophages participate in erythropoiesis [57]. Another study demonstrated how CD99 isoforms affect CD1a expression during the in vitro generation of human CD14+-derived dendritic cells (iDCs). The CD1 family of molecules ensures the presentation of lipid and glycolipid antigens from dendritic cells to T cells. The results indicated that CD99 type I downregulates CD1a mRNA and protein expression by upregulating ATF-2 phosphorylation and CREB1 expression. In contrast, CD99 type II counters CD99 type I’s down-modulatory signal, allowing CD1a re-expression. Understanding CD99 splice variants will improve the understanding of the transcriptional regulations affecting the ratio of CD1a-high/CD1a-low dendritic cells. Since dendritic cells are critical in T-cell-mediated anticancer immune responses, the modulation of CD99 signaling could significantly impact immune responses [58] (Figure 3). In addition, CD99 has been reported to be involved in regulating neutrophilic-mediated immune functions, mainly migration (see below).

### 4.4. The Role of CD99 in Cell Adhesion and Diapedesis

CD99 can act as an adhesion molecule, and CD99 engagement has been shown to induce the upregulation of the LFA-1/ICAM-1 interaction, prompting B and T-cell homotypic adhesion [23,37] or neutrophil arrest in venules [35]. CD99 has also been demonstrated to modulate integrin binding to Vascular cell adhesion protein 1 (VCAM-1), enhancing T-cell adhesion to the endothelium [59]. CD99 also regulated the adhesion of glioma cells to laminin [15].

CD99 has been shown to function downstream of Platelet endothelial cell adhesion molecule (PECAM-1), another critical molecule involved in the transendothelial migration (TEM) [60,61,62] of leukocytes and acting through homophilic interactions between CD99 on the leukocytes and CD99 on the endothelial cells [61]. It forms a signaling complex with soluble adenyl cyclase, protein kinase A (PKA), and ezrin [63]. Blocking both CD99 and PECAM-1 resulted in additive effects, suggesting that the two molecules operate at distinct steps [60,64], and demonstrated that both CD99 and CD99L2 mediate neutrophil diapedesis even under inflammatory conditions that bypass the need for PECAM-1, Intercellular Adhesion Molecule 2 (ICAM-2), and Junctional adhesion molecule A (JAM-A), suggesting that the latter participate in a pathway functioning independently of the one supported by CD99 and CD99L2. More detailed analysis revealed that neutrophils accumulated in inflamed tissues between the endothelium and the basement membrane [63,64,65].

Some studies have suggested the activation of proteases to enhance neutrophil motility within the basement membrane [64]. In this context, the research of Bedau et al. indicates that CD99 is cleaved by Meprin-β, a type I multidomain transmembrane metalloprotease that serves as an initiator of regulated intramembrane proteolysis of cell adhesion molecules [66]. Meprin-β cleaves CD99 at the cell surface, followed by intramembrane proteolysis by gamma-secretase. Consequently, this process leads to a decrease in cell adhesion and an increase in CD99-dependent TEM [67].

Another metalloprotease plays a role in the regulation of CD99. In chronic lymphocytic leukemia (CLL), matrix metalloproteinase-9 (MMP-9) downregulates CD99. Silencing MMP-9 results in increased levels of CD99, which serves as a target for MMP-9, contributing to CLL cell migration and adhesion. The downregulation of CD99 requires MMP-9 to bind with α4β1 integrin and involves Sp1 inactivation; this mechanism is active in CLL bone marrow [68]. CD99 is involved in a process known as “reverse transmigration” (RT), mainly observed in neutrophils but also occurring in T cells, B cells, monocytes, and dendritic cells (DC). In tissue-resident dendritic cells, this process is essential for migrating across the lymphatic endothelium, which is needed for adaptive immune responses. It may resemble diapedesis. A study on DC trafficking from the skin to lymph nodes found that this occurs through a homo-oligomeric interaction between CD31 (also named PECAM-1) at the junctions of lymphatic endothelial cells and CD99 on lymphatic vessel surfaces. This mirrors transmigration from blood vessels to interstitial spaces. This process aids Langerhans cells’ movement into lymphatic vessels, propelled by lymphatic flow to draining lymph nodes. The authors showed that interactions between TNF-alpha, CXCL12, CD31, and CD99 are necessary for dendritic cell “emigration” into the dermis during skin explantation. Blocking antibodies against CD31 and CD99 inhibited RT, highlighting their critical role in DC migration and trafficking [69] (Figure 4).

## 5. The Role of CD99 in Autoimmune Diseases

The expression and function of the CD99 molecule in immune cells may suggest its involvement in the etiopathogenesis of certain autoimmune diseases.

In a recent paper by Feng and colleagues, the authors explored the underlying mechanisms of osteogenesis in ankylosing spondylitis (AS), a chronic inflammatory disease characterized by bony overgrowth of the axial spine. This study identified a novel subcluster of early-stage neutrophils, CD99_G1, which was elevated in AS. The study demonstrated a close interaction between this cell population and Adipo-Cxcl12-abundant-reticular cells in the pathogenesis of AS. Moreover, CD99_G1 neutrophils also express various secreted proteins that contribute to abnormal osteogenesis [70]. As CD99 is involved in a process known as RT, mainly observed in neutrophils, it is worthwhile to speculate that targeting CD99 in this subcluster could help reduce the proportion of CD99_G1 neutrophils in AS patients.

A separate study found that CD99 expression was elevated in PBMCs and inflamed mucosa of patients with active Crohn’s Disease (CD) and ulcerative colitis (UC). Furthermore, in these patients, CD99 levels in the inflamed mucosa were higher than in unaffected control areas, accompanied by an increased secretion of pro-inflammatory cytokines [71]. Therefore, CD99 expression can serve as an indicator for evaluating inflammatory bowel disease activity. Given the established role of CD99 in leukocyte migration and T-cell activation, developing strategies that reduce lymphocyte diapedesis and subsequent inflammation in the IBD microenvironment using anti-CD99 may be a useful approach. Leukocyte trafficking into the central nervous system (CNS) drives the immunopathogenesis of multiple sclerosis (MS) and its model, experimental autoimmune encephalomyelitis (EAE). Winger et al. demonstrated that CD99 is essential for lymphocyte transmigration, while not impacting adhesion at the human blood–brain barrier model [72]. CD99 blockade in vivo improved EAE and reduced CNS inflammatory infiltrates, including dendritic cells, B-cells, and CD4+ and CD8+ T-cells. Anti-CD99 therapy was effective when given after symptom onset and prevented relapse when administered after symptom recurrence. These findings highlight CD99’s role in CNS autoimmunity and its potential as a novel therapeutic target for neuroinflammation [72].

In the study by Samus et al., CD99L2 was identified as the first endothelial adhesion receptor that facilitates the transmigration of leukocytes across the endothelial basement membrane of the blood–brai barrier (BBB). This underscoring the significance of this receptor for the development of EAE. Therefore, CD99L2 may be a target for developing novel therapeutic strategies to combat neuroinflammatory diseases [73].

All these findings suggest that CD99 is a promising target for treating autoimmune diseases. Notably, cladribine showed significant efficacy against MS in a large, randomized study, reinforcing our findings that inhibiting CD99 function could be a key mechanism behind the therapeutic effects of cladribine observed in patients with MS [74].

## 6. CD99 Functions in Tumors

While alterations in CD99 expression have been observed across a wide range of neoplastic human tissues, the precise relationship between CD99 expression and the development of human cancers remains somewhat contentious, often exhibiting opposing functions based on the cellular context. Significant CD99 expression has been noted in EWS and various forms of leukemia, including ALL [14], AML, and stem cells associated with MDS [11]. CD99 is a marker for cancer stem cells and stands out as a promising therapeutic target within these malignancies. Conversely, in another subset of tumors, CD99 expression is under-expressed compared to the normal counterpart, with downregulation appearing to be a prerequisite for cellular transformation. The exact mechanism by which CD99 functions as a tumor promoter or suppressor in different cancer types has not been completely elucidated. As highlighted in our review, certain tumors, such as EWS, are characterized by high CD99 expression. The consistent high-level expression of CD99, coupled with an EWS gene rearrangement involving *FLI1*, *ERG*, or, in rare cases, other ETS genes, serves as hallmarks of Ewing sarcoma [75]. A functional relationship exists between these factors, although it remains unclear. The introduction of *EWS-FLI1* into neuroblastoma, rhabdomyosarcoma, or mesenchymal stem cells activates CD99 expression [76,77]. However, silencing *EWS-FLI1* in Ewing sarcoma cells does not affect high-level CD99 expression [78]. The prevailing notion is that the presence of CD99 enhances the oncogenic potential of *EWS-FLI1*; however, CD99 also plays an active role in maintaining tumor malignancy. Ewing sarcoma cells lacking CD99 but expressing EWS-FLI1 show significantly inhibited growth, migration, and metastasis, suggesting that CD99 expression is vital for malignancy. Mechanistically, high CD99 levels may promote growth and cell dissemination by influencing (1) Potassium Channel Modulatory Factor 1 (KCMF1)’s inhibition of migration, which affects 14-3-3σ protein function and stabilizes MAPK signaling [79]; and (2) a general suppression of PI3K/AKT [38,80] signaling and stabilization of MAPK signaling in CD99-deprived cells following CD99 ligation. Notably, RAS/MAPK pathway activation continues in CD99-deprived Ewing sarcoma cells, with transient activation promoting proliferation and prolonged activation leading to neural differentiation. The loss of CD99 prolongs ERK1/2 phosphorylation, which is crucial for neurodevelopment and differentiation. Furthermore, CD99 silencing increases oncosuppressor miR34a, reducing Notch 1 and NF-κB signaling [81]. Thus, CD99 underpins the Ewing sarcoma oncogenic phenotype by interfering with differentiation while enhancing growth and migration

In hematopoietic tumors, CD99 is mainly expressed in T-ALL [14], early B-cell lymphoblastic lymphomas [82], immunophenotypic AML [11], and the stem cells of MDS [11]. It is often associated with disease-initiating stem cells in MDS and AML, differentiating between leukemic stem cells and normal hematopoietic stem cells in AML. This suggests that CD99 plays a role in leukemogenesis [11] In AML cells, CD99 levels increase with the Fms-like tyrosine kinase 3 internal tandem duplication (*FLT3-ITD*) mutation, which is found in approximately 30% of patients and is linked to disease progression as well as a higher risk of relapse [83,84]. This mutation occurs early in the LSC, which sustains the leukemic reservoir of the disease. Furthermore, it has been reported that the *FLT3-ITD* mutation causes tumor immune escape by upregulating CD47, which inhibits macrophage phagocytosis. Other studies have indicated that CD99 can modulate protein synthesis and simultaneously affect self-renewal, potentially contributing to the clonal expansion of HSCs and LSCs, possibly leading to AML. Additionally, as CD99 is involved in transendothelial migration, this function could impact in the dissemination of leukemic cells.

In GBM and malignant gliomas, CD99 expression is higher in undifferentiated tumors, affecting actin dynamics and motility. This links to reduced Rac and increased Rho activity, leading to more amoeboid than mesenchymal cells, highlighting CD99’s role in the amoeboid–mesenchymal transition during glioma migration. CD99 overexpression also correlates with tumor hypoxia, angiogenesis, epithelial–mesenchymal transition, metabolic reprogramming, and an M2 macrophage-dominated immunosuppressive microenvironment, aiding high-CD99 tumor progression.

CD99 is low in tumors like Hodgkin’s lymphomas, osteosarcomas, pancreatic tumors, gallbladder and gastric carcinomas, and certain pulmonary neuroendocrine tumors, while present in normal tissues. Decreased CD99 expression in gastric adenocarcinoma and pulmonary carcinoid tumors indicates poor survival and higher metastasis risk.

CD99 expression in osteosarcoma and stomach cancer reduces proliferation, migration, and metastasis while increasing differentiation. Downregulation leads to loss of normal morphology in Hodgkin’s disease, while upregulation induces terminal B cell differentiation. CD99 downregulation is crucial for tumor progression, impacting cellular identity and adhesion, which increases migration and invasion. Causes of CD99 downregulation include promoter methylation, loss of heterozygosity (LOH), and reduced SP1 expression, a key positive regulator of CD99. CD99 mutations, posttranslational modifications (like miRNA regulation), and other transcription factor abnormalities may also play a role. However, only sporadic studies are available.

CD99 influences cell migration and invasion by regulating the cytoskeleton, actin remodeling, and metalloproteinase expression, notably MMP-2 [85]. In osteosarcoma, CD99 transfection inhibits tumor metastasis by suppressing c-Src and ROCK2 activities [86]. It forms stable complexes with caveolin-1, maintaining c-Src in its inactive state [87]. Inhibiting c-Src reduces ROCK2 and ezrin levels, which connects the actin cytoskeleton to the extracellular matrix [88]. Conversely, N-cadherin and β-catenin anchor to the plasma membrane, enhancing cell adhesion and interactions. Re-expressing CD99type I strengthens cell contacts and counteracts ezrin’s promigratory effects [89]. Additionally, forced CD99type I expression downregulates actin remodeling and invasion genes like *ACTR2* and *ARPC1A*, while enhancing MAPK/ERK signaling and activated ERK recruitment, potentially increasing RUNX2 and BMP-SMAD-AP1 activity, promoting osteoblastogenesis [90].

The varying roles of CD99 across different tumors could stem not just from the cancer types but also from the differing expression levels of its two splicing isoforms (type I and type II). These isoforms are expressed in specific cell types and lead to distinct CD99 functions. They may produce different phenotypes in cancer cells, as shown in functional studies involving osteosarcoma and AML [13,86] exerting opposing effects on vital biological processes, such as migration/invasion, growth under anchorage-independent conditions, differentiation, and disease progression [91].

This section will present an updated review of the role of CD99 in tumors that overexpress this protein, focusing on the biological role of CD99 in these tumors and its interaction with the TME, as well as the implications for therapeutic strategies.

### 6.1. Leukemia

#### 6.1.1. Acute Myeloid Leukemia

CD99 is found to be overexpressed on aberrant hematopoietic stem cells (HSCs) at diagnosis or during relapse in AML, showing a positive correlation with prognosis in high-risk MDS. Indeed, the expression of CD99 enables the differentiation of leukemic stem cells (LSCs) from functionally normal HSCs in AML. Furthermore, in functional xenograft assays, NSG mice engrafted with more immature CD99-positive CD34+CD38− AML LSCs rapidly developed fatal myeloid leukemia, whereas engraftment with CD34+CD38− CD99-negative cells resulted in normal lympho-myeloid development. This confirms that CD99 plays a functional role in leukemogenesis [11]. In AML cells, CD99 expression is increased with the presence of *FLT3-ITD* mutation, which occurs in nearly 30% of patients and is linked to disease progression and higher risk of relapse [83,84]. By analyzing a substantial number of primary AML *FLT3-ITDmut* patient specimens, Angelini and colleagues discovered that the *FLT3-ITD* load was enriched in CD34/CD123/CD25/CD99+ cells compared to CD34+ progenitors (CD123+/−, CD25−, CD99low/−). *FLT3-ITD* mutations arise early in CD34/CD123/CD25/CD99+ LSCs, representing the leukemic reservoir that drives disease relapse [92]. In addition, *FLT3-ITD* induces immune escape in AML via upregulating CD47 expression and decreased phagocytic ability of macrophages [93].

Thus, CD99 may represent a therapeutic target in *FLT3*-mutated AML to eliminate chemoresistant LSCs that exhibit a robust ability for proliferation and self-renewal. This association between CD99 expression and chemoresistance in refractory and early relapsed AML patients was further validated through an integrated single-cell RNA sequencing (scRNA-seq) analysis of bone marrow samples from AML patients. This analysis demonstrated an enrichment of CD99+ CD49d+ CD52+ Galectin-1+ quiescent stem-like cells (QSCs) among CD45dim side-scatter (SSC)low blasts at diagnosis in resistant and refractory/early relapsed patients, as well as in chemotherapy-resistant residual cells. These findings suggest that CD99 may play a role in immune evasion and the maintenance of leukemic stemness [94].

Indeed, Huang et al. demonstrated that CD99 is essential for the self-renewal of proliferating HSCs and LSCs by negatively regulating protein synthesis. The absence of CD99 in HSCs and LSCs leads to increased protein synthesis and impaired self-renewal, which can be restored by inhibiting translation. These findings emphasize CD99’s key role in regulating protein synthesis, which may contribute to the clonal expansion of HSCs and LSCs, potentially leading to AML [95]. Another mechanism through which CD99 contributes to the aggressiveness of AML may include its ability to enhance transendothelial migration and mobilization of malignant leukemia cells [96].

As previously mentioned, CD99 is overexpressed on disease-initiating cells in most AML cases, especially in the more immature CD34+/CD38− cells often linked to chemoresistance and a reduced ability to trigger immune responses [97]. Notably, this feature is not seen with other previously identified LSC markers, highlighting CD99 as a possible new therapeutic target in AML. Moreover, the anti-CD99 treatment effectively induces the death of AML LSCs and blasts while leaving normal HSCs unaffected [11,84].

The specific mechanism by which CD99 ligation correlates with cell death in tumor stem cells and blasts while sparing normal HSCs remains somewhat unclear. Mechanistically, it involves the activation and modulation of the SRC and ERK pathways, a decrease in MDM2 levels, and increased p53 activation that triggers both extrinsic and intrinsic apoptotic pathways [11,13,84]. These mechanisms are similar to those observed in Ewing sarcoma cells after the engagement of CD99 with monoclonal antibodies [98]. Additionally, in AML cell lines and AML blasts from patients, CD99 targeting affects cell metabolism by reducing glycolysis and mitochondrial respiration, paving the way for combinations with metabolically active drugs [84]. As reported above, CD99 may play a role in activation-related exocytic processes. CD99’s involvement with the post-Golgi trafficking system regulates the transport of various proteins, including MHC and HLA class I molecules, to the plasma membrane [44,45]. This may represent a mechanism through which tumor cells exploit CD99 expression for immune response evasion. In FLT3-ITD mutated AML cells, CD99 colocalizes with FLT3-ITD in the perinuclear region. The engagement of CD99 with monoclonal antibody 3.2-3 (in preclinical development at Diatheva srl) triggers FLT3-ITD mobilization from the perinuclear region to the cell membrane [84], similar to what is observed upon treatment with FLT3-ITD inhibitors quizartinib (AC220) and midostaurin (PKC412) [99].

The increase in FLT3-ITD levels at the plasma membrane due to TK inhibitors’ induced translocation can be leveraged in immunotherapy to boost T-cell-mediated cytotoxicity (TCM) driven by a bispecific FLT3-CD3 antibody [100].

Similarly, CD99 regulates FLT3-ITD localization, suggesting its potential in immunotherapy for acute myeloid leukemia. Recently, CD99 has been leveraged to create scFvs-based nanoparticles targeting both CD99 and FLT3. This dual-targeting shows promising antileukemic effects in AML cell lines and primary blasts with the *FLT3-ITD* mutation, downregulating adipogenesis, fatty acid metabolism, and IL6/JAK/STAT3 and PI3K/AKT/MTOR pathways [101]. This research team also examined the T-cell response after treating AML cell lines with α-CD99-A192 nanoparticles. This treatment induces cell aggregation and proliferation while upregulating inflammation and immune response-related genes. Additionally, it downregulates genes linked to T-cell exhaustion, like Programmed cell death protein 1 (PD-1) and Cytotoxic T-Lymphocyte Antigen 4 (CTLA-4), resulting in a stronger cytotoxic response to leukemic cells [102].

Regarding the expression of CD99 isoforms in AML, transcriptomic dataset analyses from AML patients reveal that the CD99 type II isoform is the most abundantly expressed. Induced overexpression of CD99 type I in AML cell lines has shown that this isoform promotes leukemia growth. However, this proliferation is serum-induced and not sustained over the long term. Moreover, overexpression of CD99 type I leads to increased ROS levels and apoptosis, suggesting that the CD99 type II isoform may support AML cells’ survival [13].

#### 6.1.2. Acute Lymphoblastic Leukemia

CD99 is highly expressed in newly diagnosed T-ALL compared to normal hematopoietic stem cells and is also used as a marker for detecting minimal residual disease (MRD) [14,103,104]. Like AML, in T-ALL, ligation of CD99 by antibodies induces the death of malignant T cells but does not affect mature peripheral T cells. Several anti-CD99 mAbs have been studied for their direct effects on T-ALL cells and found to exert different outcomes based on their recognition epitope in CD99. Some of these mAbs (DN16, 0662, Ad20) can induce death in T-ALL cells, while others, such as mAb 12E7 and mAb D44 mIgM, fail to induce apoptosis in Jurkat cells and immature thymocytes. Besides the binding epitope, antibody valency in the ligation of CD99 also appears to influence the induction of cytotoxicity, and the co-expression of CD99 long and short isoforms present on T-ALL cells is involved in the activation of apoptotic signaling. For a complete review of CD99 as an antibody target in T-ALL, please refer to Kotemul K et al. [105]. Recently, CD99 has gained attention as a potential target for Chimeric Antigen Receptor T-Cell (CAR-T) therapy for T-ALL. Specifically, treating Cell-Derived Xenografts (CDXs) and Patient-Derived Xenografts (PDXs) models developed from T-ALL cell lines and patient samples with 12E7 anti-CD99 CAR T cells has shown delayed leukemia progression and the elimination of infiltrating leukemia cells in the spleen and bone marrow. Additionally, the selective cytotoxic activity of anti-CD99 CAR T cells has been observed in several preclinical cellular models of AML and solid tumors, underscoring the potential of CD99 in the design of immunotherapeutic strategies [106]. An innovative strategy to reduce fratricide, a major limitation in CAR-T therapy for ALL, has been developed by Anqi Ren et al. By targeting CD99, they discovered that directly transducing the T-ALL Jurkat cell line with the anti-CD99 single-chain variable fragment (scFv) 12E7 prompted the self-elimination of tumor cells, providing a promising solution to enhance CAR-T efficacy in T-ALL [107].

In addition to the direct effects of induced cytotoxicity on T-ALL cells, antibodies targeting CD99 have been shown to disrupt the adhesion between leukemia cells and meningeal cells by inducing matrix metalloprotease activity. Interestingly, studies have demonstrated direct interactions between CD99 expressed in leukemia cells and meningeal cells in the central nervous system (CNS), enhancing leukemia chemotherapy resistance by affecting the balance of leukemia apoptosis and the cell cycle. The disruption of the CD99-CD99 interaction has been shown to restore the sensitivity of leukemia cells to chemotherapy. This research identifies a mechanism that regulates critical intercellular interactions within the CNS leukemia niche and supports the targeting of CD99 as a potential approach for overcoming meningeal-mediated leukemia chemoresistance [108] (Figure 5).

### 6.2. Malignant Glioma/Glioblastoma

The expression of CD99 in glioblastoma (GBM) has been the focus of extensive research, with several studies investigating its role in various aspects of the disease [15,16,109,110,111,112]. CD99 expression levels are found to be elevated in GBM compared to normal brain tissue or lower-grade gliomas [15,16,109,112]. Only the long isoform of CD99 is expressed in GBM cells. However, the prognostic significance of CD99 expression remains under-explored, with no clear evidence of its association with overall survival (OS) [16,113]. The prognostic impact of CD99 expression emerged only when considering gliomas of all grades [112]. The role of CD99 has been found to correlate with cell invasiveness and migration, underscoring its potential role in the tendency of GBM to microscopically infiltrate adjacent normal brain tissue, leading to frequent residual microscopic disease even after radical surgery. Seol HJ and colleagues confirmed that gliomas exhibit elevated CD99 expression compared to non-neoplastic brain tissue and that silencing CD99 in glioma cells reduces migration and invasiveness without affecting cell viability or proliferation [15]. Conversely, the ectopic expression of CD99 in glioma cells resulted in increased cell migration and invasiveness. This finding was further confirmed in an orthotopic brain tumor model, where CD99 expression significantly enhanced the tumor’s invasiveness [15]. This phenotype was associated with decreased Rac activity and increased Rho activity along with a significantly increased proportion of amoeboid cells to mesenchymal cells, suggesting the involvement of CD99 in amoeboid–mesenchymal transition in glioma migration [15]. Consequently, CD99 emerges as a promising future target to impede migration and invasion. A separate study has indicated a correlation between the highest levels of CD99 expression in GBM and its relationship with larger, multilobular tumor extensions and increased migratory activity capacity [16]. CD99 type I expression was also found to be higher in the classical and undifferentiated mesenchymal GBM subtypes compared to the pro-neural subtype, indicating that it may be associated with the Proneural–Mesenchymal transition (PMT). In gliomas, therapeutic pressure induces PMT, resulting in increased tumor initiation and recurrence potential, as well as the development of resistance to therapies [114].

It was also linked to the expression of genes involved in actin dynamics, supporting the formation of focal adhesions, lamellipodia, and filopodia on the leading edge of moving cells, maintaining the association of the antigen with motility [109]. A bioinformatic analysis of multiple glioma datasets identified CD99 as a biomarker that characterizes the adaptive response. Shang and colleagues demonstrated an association between CD99 overexpression and tumor hypoxia, angiogenesis, epithelial–mesenchymal transition, metabolic reprogramming, and an immunosuppressive microenvironment dominated by M2 tumor-associated macrophages (TAMs) [112]. Since TAMs produce low levels of pro-inflammatory cytokines and lack T-cell co-stimulation factors, this leads to T-cell exclusion, facilitating tumor escape from immunosurveillance [112,115]. High CD99-expressing gliomas tolerate stress conditions like hypoxia and antitumor immunity, affecting responses to anti-angiogenic and immune checkpoint inhibitors. In responders to anti-angiogenic drugs like bevacizumab, CD99 expression decreased, while non-responders showed no significant changes. Higher CD99 levels correlated with less T-cell infiltration and more T-cell exhaustion, influenced by elevated TAMs, myeloid-derived suppressor cells (MDSCs), and regulatory T cells, potentially impacting immune checkpoint responses. Furthermore, glioma patients with high CD99 expression exhibited resistance to chemotherapy and radiotherapy, and a positive correlation was found between increased CD99 levels and poor short-term recurrence and survival [115]. Therefore, for patients with CD99-overexpressing gliomas, it is essential to consider tumor adaptiveness during treatment to prevent drug resistance, and closer clinical monitoring should be conducted to improve the effectiveness of treatment [112].

Interestingly, in GBM, CD99 has been implicated in cuproptosis, a type of copper-induced cell death, through its interaction with the vascular endothelial growth factor (VEGF) pathway [116]. This study revealed a direct relationship between CD99 and VEGFA expression and the Cuproptosis Activation Scoring (CuAS) score, showing opposite results in tissues with low CuAS. This offers a new perspective for targeting cuproptosis-related tumor cells. CD99 is highly expressed in diffuse midline gliomas (DMGs), particularly with the *H3K27M* mutation, which indicates poor prognosis. In K27M+DMG preclinical models, the anti-CD99 chimeric antibody 10D1 reduced DMG growth by inducing apoptosis. Additionally, combining 10D1 with radiation significantly improved antitumor efficacy and survival in xenograft models, supporting its clinical application in DMG treatment [117] (Figure 6).

### 6.3. Ewing Sarcoma

High expression of CD99 is a common and distinctive feature of EWS cells. It has been widely used for differential diagnosis from other types of small round cell tumors in children and adolescent–young adults (AYAs) [10,118,119] or as a potential risk factor for disease onset and progression [120]. When CD99 is knocked down in EWS cells transplanted into immunodeficient mice, it results in terminal neural differentiation, subsequently leading to reduced tumor growth, migration, and incidence of bone metastasis [79,80]. Due to this evidence, CD99 has gained attention for understanding the mechanisms related to the pathogenesis of EWS and as a potential target for targeted therapy.

Multiple lines of evidence suggest a functional relationship between the aberrant transcription factor EWS-FLI, which is the pathogenic driver of EWS, and CD99 [76,77,121,122]. CD99 enhances the oncogenic activity of EWS-FLI, as EWS-FLI consistently maintains elevated levels of CD99 expression [77,80,121]. This occurs directly at the transcriptional level via its interaction with the CD99 promoter [80,123] or posttranscriptionally through the downregulation of miRNA-30a-5p expression, which targets the 3’-UTR region of CD99, likely acting as a translational repressor [78]. A recent analysis of transcriptomic data from in vitro and in vivo models with gene silencing of *CD99* and *EWS-FLI1* reveals an increase in *HDAC* target genes, indicating that these factors influence gene expression through histone modification [124]. CD99 and EWS-FLI influence EWS cell differentiation in opposing ways. EWS-FLI fosters neural differentiation, whereas CD99 inhibits it [80]. When EWS-FLI and CD99 are expressed together, as in EWS, they create a combined effect in malignant cells that enhances the expression of certain neural traits while preserving cell growth potential. Silencing *CD99* in human EWS cell lines induces prolonged nuclear ERK1/2 phosphorylation [80], which appears essential for redirecting the biological functions of ERK1/2 toward neural development and differentiation [125] while decreasing AKT and NF-κB signaling [80,81]. This regulatory mechanism occurs indirectly and involves miR-34a as an inhibitor of Notch1 and NF-κB, which is enriched in exosomes derived from CD99-silenced EWS cells [81]. Ultimately, this mechanism guides the cells toward a terminal neural differentiation state regardless of the presence of EWS-FLI. Thus, CD99 may function as a way to fine-tune the levels of transcriptional gene regulation and shift the balance in favor of cell differentiation instead of proliferation.

A recent paper demonstrated that EWS-FLI1 and CD99 simultaneously inhibit miR-214-3p function, leading to a failure to inhibit High-mobility group protein A1 (HMGA1) protein, thereby maintaining the undifferentiated state and tumor aggressiveness [126]. Other studies suggest that CD99 in EWS has a regulatory role in exosome trafficking and their miRNA cargo, promoting the upregulation of Jun/Fos and, consequently, AP-1 activity through miR-199a-3p, which is enriched in CD99-negative exosomes from CD99-depleted EWS cell lines [127]. Similarly, CD99 modulation can influence extracellular vesicle cargo, as De Feo et al. demonstrated recently. In fact, they showed that a decrease in CD99 leads to significant changes in the proteomic profiles of EWS cells and their extracellular vesicles. These changes involve cell migration and immune modulation proteins, with CD99-silenced cells and their extracellular vesicles (EVs) characterized by a migration-suppressive, pro-immunostimulatory proteomic profile [128].

EVs produced by EWS cells express CD99 at the plasma membrane, suggesting that this detection could be considered a biomarker for diagnosing EWS through liquid biopsy [129]. Similarly, CD99 represents a crucial target for identifying and isolating circulating tumor cells (CTCs), thereby aiding in the detection of potential micro metastases during the follow-up of EWS patients in the late stages of tumor progression [130].

The high expression of CD99 on the cell surface and its crucial role in sustaining tumor malignancy are the two primary reasons why it has garnered significant interest over the years for developing new targeted therapies using mAbs for tumors that overexpress CD99. Most of these antibodies are known to trigger cell death signals and prevent cell migration. In EWS, CD99 engagement induces tumor cell death through a unique, non-conventional, caspase-independent programmed cell death involving zyxin and actin as mediators of this process [131]. Specifically, CD99 engagement promotes the translocation of zyxin into the nucleus of EWS cells, inhibiting the transcriptional activity of *GLI1*, a target of EWS-FLI1. This alters its associated target genes, which regulate cell proliferation, stemness, and migration [132].

Additionally, another study found that antibody-mediated engagement triggers caveolin-1-dependent endocytosis, activating the IGF1-R/Ras/Rac1 signaling pathway. This leads to an accumulation of fluid-filled vacuoles that lack lysosomal enzymes and acidic pH, resulting from aberrant macropinocytosis, and induces cell death via a non-apoptotic mechanism like methuosis [133]. Furthermore, anti-CD99 antibodies show additive and synergistic effects when paired with conventional drugs, such as doxorubicin or vincristine [98,134], and they also maintain their effectiveness against chemoresistant EWS cells [133].

The impact of anti-CD99 antibody treatment is more pronounced in malignant cells that express high levels of CD99, driven by the reactivation of p53, resulting from the CD99-mediated degradation of MDM2 [98]. From a safety perspective, the engagement of CD99 with anti-CD99 mAbs does not induce p53 reactivation or RAS induction in normal cells. Similarly, no macropinocytosis or general toxicity was observed in CD34+ hematopoietic stem cells, mesenchymal stem cells, or macrophages expressing high levels of CD99 [8,133,134]. This evidence suggests that CD99-induced cell death primarily occurs in tumors with abnormal genetic backgrounds, such as EWS and leukemia, thereby providing selectivity.

A human monospecific bivalent single-chain fragment variable (scFv) diabody (dAbd C7 in development at Diatheva) targeting CD99 by recognizing a different epitope was developed [135] and characterized recently, with encouraging results similar to those obtained with murine antibodies [98] but with the advantage of preventing a potential human anti-mouse antibody (HAMA) response and enhanced penetration into the tumor mass.

Researchers are exploring small molecules that mimic CD99 engagement effects as potential EWS therapies. The FDA-approved nucleoside analogs clofarabine and cladribine directly bind to the extracellular part of CD99, blocking its dimerization and associated signaling. This signaling involves interactions between CD99, cyclophilin A, and PKARIIα which have been previously described as CD99 partners to regulate leukocyte TEM [63,136]. Inhibiting CD99 with these molecules induces a strong cytotoxic response in Ewing sarcoma cells, decreases motility by reducing ROCK2 expression, and hinders anchorage-independent growth. Additionally, xenograft models in mice indicate that clofarabine and cladribine treatments inhibit tumor growth in vivo [137]. Mechanistically, clofarabine exhibits similarities to anti-CD99 antibodies, leading to the rapid induction of ERK1/2. This ERK1/2 activation leads to MSK1/2 phosphorylation, which activates CREB and can cause a fast response to various cellular responses related to cell growth and differentiation. However, even though these pathways are not directly involved in triggering cell death, they are independent of the anti-CD99 antibodies used [138]. To extend the therapeutic window of these two nucleoside analogs, membrane-impermeable variants, BK60106 and BK50164, have been developed. These variants can act extracellularly on CD99 to induce cell death in EWS cells without affecting DNA synthesis, exhibiting greater specificity for targeting CD99 expressed in cancer cells while demonstrating much lower toxicity towards normal proliferating cells [139].

In recent years, there has been a significant increase in treatment proposals for EWS using immunotherapeutic strategies aimed at counteracting the immunosuppressive TME [140]. However, the potential involvement of CD99 remains underestimated. Recently, Manara et al. showed that anti-CD99 antibody treatment creates a virtuous cycle that delivers intrinsic cell death signals to EWS cells; promotes tumor cell phagocytosis by macrophages that, like EWS cells, express CD99 at high levels; and induces the reprogramming of M0 macrophages to M1-like macrophages, which are pro-inflammatory and typically associated with tumor regression. The enhanced tumor cell phagocytosis directly relates to the inhibition of CD99-CD99 interactions between EWS cells and macrophages. It is not triggered by Fc effector functions like ADCC, as demonstrated using the anti-CD99 diabody C7 dAbd, which lacks the Fc portion. In EWS xenografts, anti-CD99 antibody treatment resulted in significant tumor regression and the recruitment of pro-inflammatory macrophages. This suggests that CD99 may also enhance the antitumor activity of macrophages in vivo [8]. O’Neil and colleagues described similar evidence showing that anti-CD99 antibody treatment induced EWS tumor regression and the recruitment of macrophages. Furthermore, the authors demonstrated that co-incubation with an anti-PILRalpha antibody leads to the reactivation of previously dormant macrophages, likely due to the interrupted binding of Ewing CD99 to the PILRalpha macrophage inhibitory receptor [9]. High levels of CD47 and low levels of membrane-exposed CALR are associated with a dual mechanism utilized by EWS cells and other tumors to evade the macrophage response. Responses similar to those caused by CD99 engagement on CD47 and CALR levels [8] have recently been achieved by combining doxorubicin and an anti-CD47 antibody [141]. In this context, treatment with anti-CD99 mAbs emerges as a novel immunotherapeutic tool capable of expanding the therapeutic window, yielding results comparable to those obtained with non-selective chemotherapeutic agents (Figure 7).

## 7. Conclusions

CD99 is a molecule with unexplored functions in tumor biology and the tumor microenvironment. Most studies involving CD99 have historically explored the expression of this antigen in tumors and its potential use as a tumor biomarker for diagnosis and follow-up. Researchers have only begun to investigate the possible use of CD99 as a therapeutic target in recent years, leveraging its engagement with monoclonal antibodies to induce cell death in tumor cells that highly express this protein.

Recent insights highlight CD99’s role in the communication between tumors and immune cells and with cells within the tumor niche. CD99 has been shown to play a role in immune escape, invasiveness, migration, and the adaptability of tumor cells exposed to various stress conditions, including treatment with certain therapeutics (Figure 8).

This opens new frontiers in targeting this antigen and potential combinations of anti-CD99 with other therapies, such as immunomodulatory drugs or immune checkpoint inhibitors.

Nevertheless, several critical issues remain to be resolved:Information on the expression of the two isoforms in immune and tumor cells is still lacking, as is a shortage of specific antibodies capable of distinguishing between them. A clearer definition of these isoforms’ expression across cells could enhance our understanding of their function.The interplay between tumor cells and the immune microenvironment mediated by CD99 is a subject of great interest. A comprehensive understanding of how the presence or absence of CD99 may impact tumor infiltration and the communication between tumor cells and normal host cells is essential for accurately defining effective innovative therapeutic strategies.The potential to enhance the tumor microenvironment by triggering CD99 with antibodies or compounds that mimic CD99 in immune-cold tumors could provide new therapeutic perspectives for immune-evasive tumors. However, further studies are necessary to fully comprehend the benefits and minimize any collateral effects that this modulation may cause.

## Figures and Tables

**Figure 1 biomolecules-15-00632-f001:**
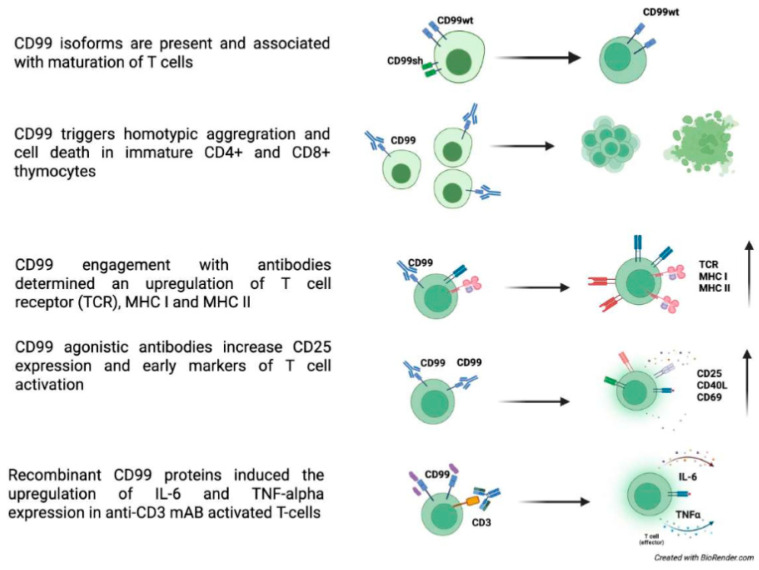
CD99 roles in T cells [3,4,5,6,23,37,39,40,41,44].

**Figure 2 biomolecules-15-00632-f002:**
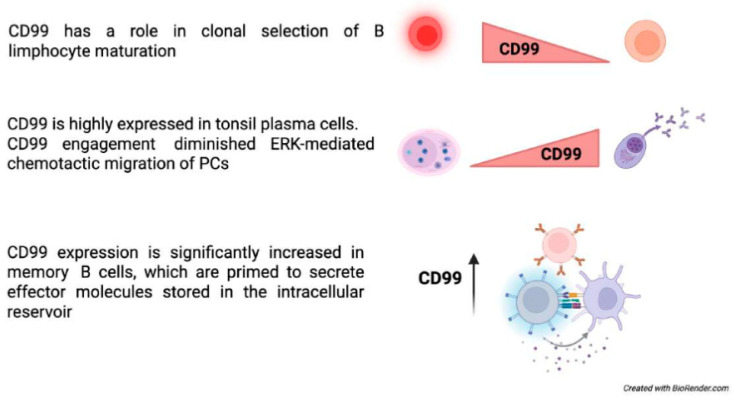
CD99 functions in B-cells [48,49,54,55,56].

**Figure 3 biomolecules-15-00632-f003:**
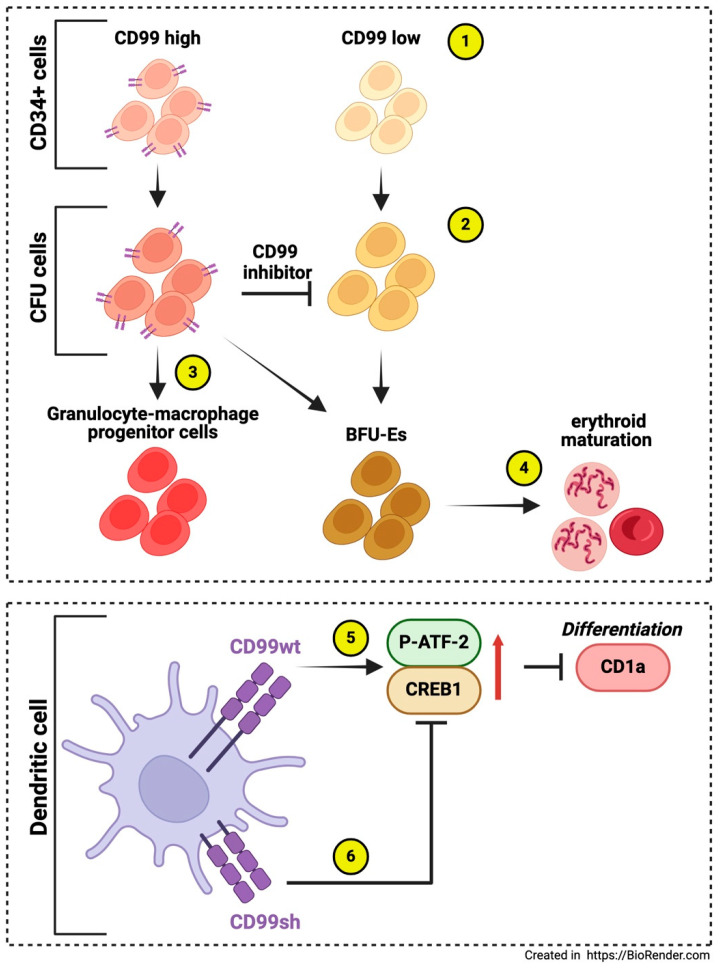
The roles of CD99 in other immune cells. CD34+ cells exhibit subpopulations with varying CD99 expression (1), like CFU-E cells (2). These populations include a CD99high subcluster that retains the potential for differentiation into granulocytes, macrophages, and erythroid cells (3). Inhibition of CD99 reduces the proliferation of pluripotent stem cell-derived cells and promotes erythroid maturation, primarily in the CD99 low subcluster (4) [57]. In dendritic cell differentiation in vitro, CD99wt downregulates CD1a mRNA and protein expression by upregulating ATF-2 phosphorylation and CREB1 expression (5), while CD99sh counteracts the down-modulatory signal of CD99wt, permitting the re-expression of CD1a (6) [58].

**Figure 4 biomolecules-15-00632-f004:**
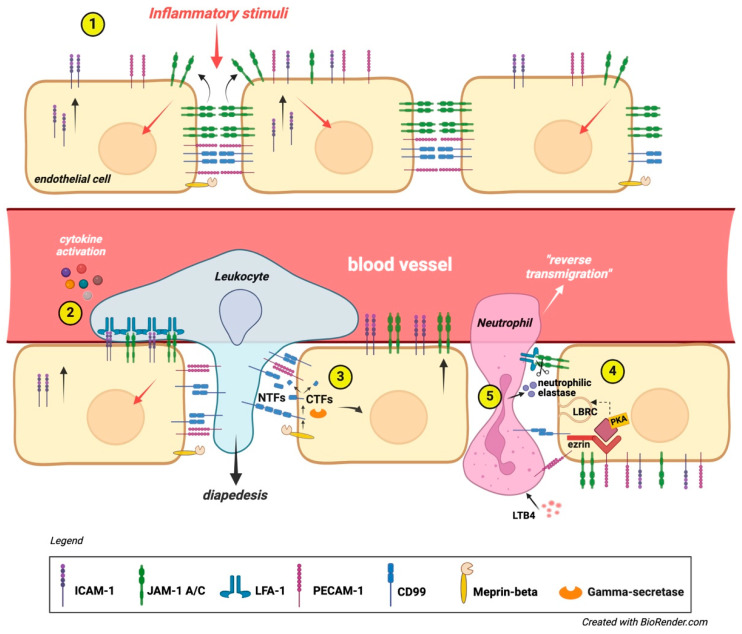
CD99 role in diapedesis. (1) An inflammatory stimulus triggers ICAM-1 expression on the luminal surface of the vascular endothelium. This signal also influences the junctional placement of JAM-1 at inter endothelial connections, while the lateral positioning of CD99 and PECAM-1 remains unchanged. (2) Leukocytes transmigrate across the vascular endothelium using JAM-1, PECAM-1, and CD99. Chemokines activate leukocyte integrins, resulting in LFA-1 binding to vascular ICAM-1 and JAM-1. This is followed by sequential homophilic interactions of PECAM-1 and CD99, facilitating the trans-endothelial migration of leukocytes [23,35,37]. (3) In some cases, the activation of proteases could mediate neutrophil TEM across the basal membrane. CD99 is cleaved by meprin-beta, and the remaining C-terminal fragments (CTFs) undergo further cleavage by gamma-secretase. The shedding of CD99 from the endothelium weakens the homotypic ligation with leukocytes, promoting TEM [67]. (4) Under resting conditions, ezrin, soluble adenylyl cyclase (sAC), and PKA form a signaling complex at intercellular junctions. During TEM, leukocytes mobilize lateral border recycling compartment (LBRC) vesicles to the endothelial cell junctional plasma membrane, increasing the surface area. This process is initiated by PECAM-1 and CD99, with CD99 signaling through ezrin-associated sAC and PKA. Leukocyte transmigration becomes irreversible due to junctional adhesion molecule C (JAM-C); (5) loss of function or reduced JAM-C expression exacerbates reverse transmigration from the abluminal to luminal side of the endothelium. In ischemia–reperfusion injury, leukotriene B4 (LTB4) prompts neutrophils to secrete elastase, degrading JAM-C and increasing reverse transmigration [63].

**Figure 5 biomolecules-15-00632-f005:**
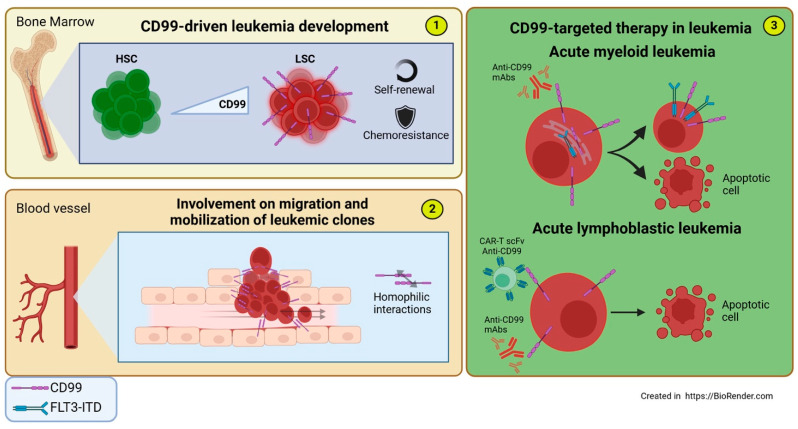
Involvement of CD99 in acute Leukemia and CD99-targeted therapies. (1) The expression of CD99 drives the differentiation from functionally normal HSCs to AML LSCs, conferring self-renewal capacity, chemoresistance, and the maintenance of a reservoir for the formation of new leukemic clones. (2) CD99 promotes the transendothelial migration and mobilization of malignant leukemia cells through homophilic CD99-CD99 interactions between leukemia cells and endothelial cells. (3) CD99-targeted therapy in leukemia: (**Top**) Anti-CD99 mAb treatment in AML cells promotes FLT3-ITD mobilization from the perinuclear region—where it co-localizes with CD99—to the cell membrane, triggering intrinsic and extrinsic apoptotic responses. (**Bottom**) In T-ALL cells, treatment with anti-CD99 mAbs or CAR-T cells engineered with scFv anti-CD99 induces cytotoxicity.

**Figure 6 biomolecules-15-00632-f006:**
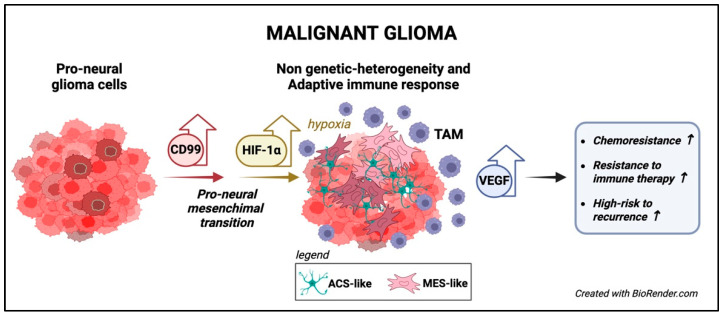
The functions of CD99 in malignant gliomas. CD99 is highly expressed in classical undifferentiated GBM and is associated with the PMT [114]. CD99 overexpression correlates with tumor hypoxia, angiogenesis, metabolic reprogramming, and immunosuppressive microenvironment characterized by M2 TAM [112,115]. High CD99 expression is associated with resistance to therapy, including immune therapy, and a higher recurrence risk [112,115].

**Figure 7 biomolecules-15-00632-f007:**
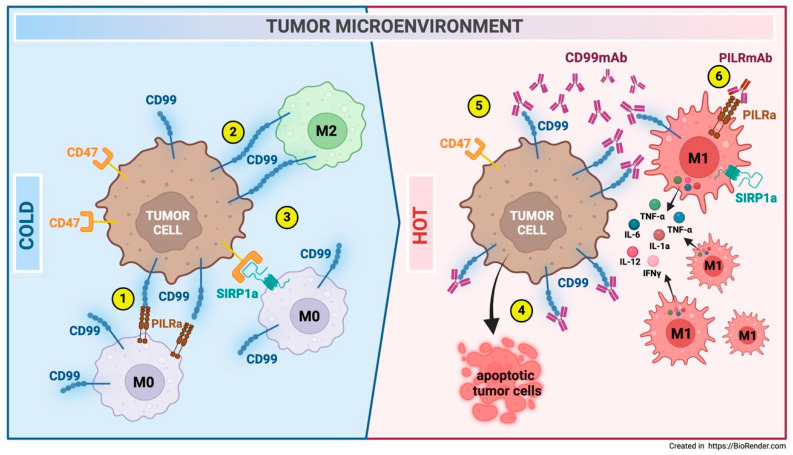
Modulation of the EWS TME following anti-CD99 antibody treatment. An immune-cold microenvironment with M0 and M2 macrophages characterizes EWS. Macrophages interact with EWS cells via CD99-PILRa inhibitory (1) and CD99-CD99 homotypic ligations (2). These interactions suppress macrophage activity. Additionally, CD47, which is highly expressed in EWS cells, inhibits macrophage phagocytosis through SIRP1a ligation (3). Treatment with anti-CD99 antibodies induces EWS cell death (4), downregulates CD47 expression (5), and promotes macrophage recruitment and polarization toward an M1-like pro-inflammatory antitumor phenotype (6) [8]. Co-treatment with an anti-PILR1-α antibody leads to the reactivation of previously dormant macrophages, likely due to the disrupted binding of Ewing CD99 to the PILR-α macrophage inhibitory receptor (6) [9].

**Figure 8 biomolecules-15-00632-f008:**
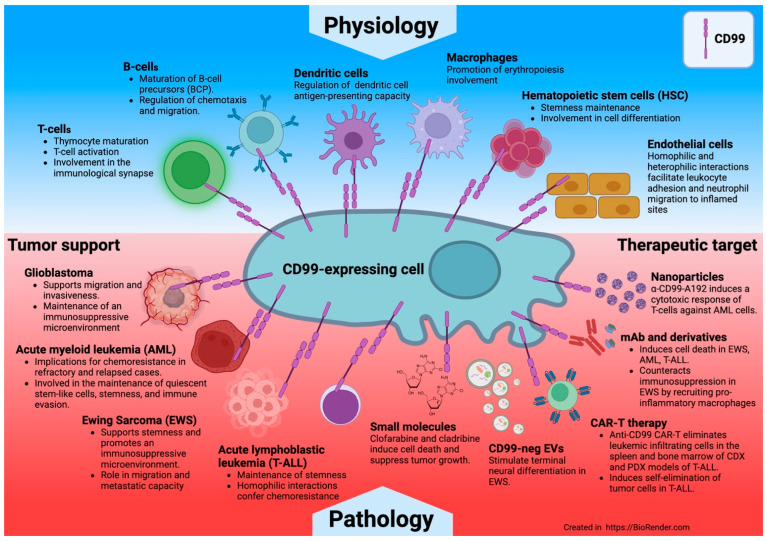
CD99 protein at the intersection of physiology and cancer.

## Data Availability

Not applicable.

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
