# Peer review of "CD99: A Key Regulator in Immune Response and Tumor Microenvironment"

_biomolecules, 2025, doi:10.3390/biom15050632_

Round 1

Reviewer 1 Report

Comments and Suggestions for Authors

The manuscript "CD99: A Key Regulator in Immune Response and Tumor Microenvironment" addresses significant aspects of CD99’s role in immunoregulation, offering insights that could benefit researchers exploring this molecule’s functions, particularly in cancer and immune responses. However, several critical points require attention to strengthen the review and enhance its scientific impact. These are outlined below:

Elaboration on CD99’s Dual Roles in Cancer:

The authors note that CD99 exhibits dual and context-dependent roles in cancer, potentially linked to its isoforms, but they defer explanation to an external reference (Reference 69) without further discussion. Given that the primary focus of this review is CD99’s role in cancer, it is insufficient to direct readers elsewhere for such a pivotal concept. The authors should provide a comprehensive explanation of the potential mechanisms underlying these dual roles, including how different isoforms may contribute to divergent outcomes, to support their conclusions and enhance the manuscript’s standalone value.

CD99 Isoforms and Immune Cell Function:

The review would benefit from a detailed discussion of how CD99 isoforms influence the function of distinct immune cell types within the tumor microenvironment. The authors should explore and clarify the contributions of these isoforms to immune cell behavior and their implications for cancer immunoregulation. This addition would strengthen the mechanistic insights and align with the manuscript’s goal of elucidating CD99’s regulatory roles.

Reviewer 2 Report

Comments and Suggestions for Authors
  1. Some sections use inconsistent terminology for the same concept (e.g., "tumor microenvironment" vs. "TME"). Standardizing terminology throughout the manuscript would improve clarity and coherence.

  2. Some references are outdated and could be supplemented with more recent studies to ensure the manuscript reflects the latest advancements in the field.

  3. The manuscript provides in-depth descriptions of certain molecular pathways, while others are only briefly mentioned. A more consistent level of detail across pathways would enhance scientific rigor.

  4. While the introduction discusses CD99’s general role, it does not immediately highlight the significance of its dual function in immune regulation and tumor progression. Adding a sentence to emphasize this distinction upfront would strengthen the introduction.

  5. The manuscript states that CD99 influences MHC class I transport and T-cell activation, but it is unclear whether this effect is direct or mediated through another signaling cascade. Could the authors clarify this point?

  6. The manuscript describes how CD99 overexpression correlates with tumor progression in some cancers, while in others, its downregulation is observed. What factors determine whether CD99 functions as a tumor promoter or suppressor in different cancer types?

  7. In the glioblastoma section, the role of CD99 in the proneural-mesenchymal transition (PMT) is discussed. Could the authors expand on whether this process presents a viable therapeutic target?

  8. The discussion on anti-CD99 antibodies highlights their effectiveness in preclinical models. Are there any ongoing or planned clinical trials investigating these therapies?

  9. It is mentioned that anti-CD99 therapies may synergize with immune checkpoint inhibitors. Has this been evaluated in combination studies, and if so, what were the findings?

  10. The section on small-molecule inhibitors like clofarabine is intriguing. Could the authors provide additional details on their mechanism of action compared to antibody-based strategies?

  11. Given that CD99 plays a role in immune cell interactions, could it also contribute to autoimmune diseases? If so, how might this impact therapeutic development?

Round 2

Reviewer 2 Report

Comments and Suggestions for Authors

We satisfied the revised manuscript.